# Evaluating word representation for hypernymy relation: with focus on Arabic

## Abstract

Hypernymy relation is one of the fundamental relations for many natural language processing and information extraction tasks. A key component of the performance of any hypernymy-related task is word representation. Traditional word embeddings capture word similarity but fall short of representing more complex lexical-semantic relationships between terms, such as hypernymy. To overcome this, recent studies have proposed hypernymy-specific representations. In this study, we conduct an evaluation of several types of word representations to determine the most effective approach for modeling hypernymy relationships in Arabic. We use an Arabic training corpus and several datasets to assess traditional embedding, hypernymy-specific embedding, and contextual embedding across several hypernymy-related tasks, including hypernymy detection. The results indicate that different embeddings have different effects on the performance. Moreover, the performance is affected by the selected datasets. This highlights that there is a need for further research to develop more robust word representation and benchmark datasets.

## 1 Introduction

Hypernymy is a lexical semantics relation that occurs between two terms in which the meaning of one is enclosed in the meaning of the other Na & Khoo (2006). Hypernym is the more general term, while hyponym is the more specific term; for example, in the sentence "cappuccino is a type of coffee" cappuccino is the hyponym, and coffee is the hypernym. Terms that share the same hypernym are called co-hyponyms Na & Khoo (2006). Hypernymy relation plays a crucial role in many Natural Language Processing (NLP) and Information Extraction (IE) applications, such as query expansion, ontology building, and machine translation. Because of its importance, several tasks in the literature are devoted to identifying hypernymy relations, some of which are: **hypernymy extraction**, which extracts hyponyms and their hypernyms from a corpus, **hypernymy detection**, which aims to distinguish hypernymy from other relations, **hypernymy directionality detection**, which aims to identify the direction of the relation, i.e., whether the general term comes first or second, **hypernymy discovery**, which aims to discover candidate hypernyms on a corpus for a query hypernym and **semantic relations classification** which aim to classify semantic relations including hypernymy. Word representation is a fundamental step in all NLP and IE tasks. Numerous types of word representation exist, starting from basic sparse and dense representations such as one-hot encoding and term matrix post-processed with singular value decomposition (SVD) to complex representations such as neural embeddings and graph embeddings. Recently, the use of neural word embeddings widespread across NLP tasks, and many tasks have adopted the use of traditional word embedding such as word2vec Mikolov et al. (2013), GloVe Pennington et al. (2014), and FastText Bojanowski et al. (2017). General word embedding can model semantic similarity and relatedness between terms. Word similarity encodes various lexico-semantic and topical relations such as synonymy, antonymy, hypernymy, co-hyponymy, and meronymy Weeds et al. (2014). Some studies have proposed hypernymy-specific representations to better model hypernymy-relation in hypernymy-related tasks. In this study, we will show the effect of different types of representations on hypernymy-related tasks, especially in Arabic. We have studied traditional word embedding, hypernymy-specific word embedding, and contextual word embedding. Our hypothesis is that the representation used will greatly impact performance in terms of the f1-score and Average Precision (AP). We have evaluated the embeddings on hypernymy detection, hypernymy directionality, and

semantic relation classification. Our results show that the effect of the used embedding is highly dependent on the testing datasets and on the vocabulary of the used lexical constraints.

## 2 RELATED WORK

The input for most NLP models is a representation of a text. The complexity of these representations has varied over time, from frequency-based representations such as Term Frequency-Inverse Document Frequency (TF-IDF) Manning et al. (2008) to contextual neural embedding such as Bidirectional Encoder Representations from Transformers (BERT) Devlin et al. (2019). The input for most hypernymy-related tasks' models is a pair of words. Studies have adopted different types of representation techniques; some have used basic frequency-based representations such as Pointwise Mutual Information (PMI), Positive-PMI (PPMI), and Singular Value Decomposition (SVD) Weeds et al. (2014); Shwartz et al. (2017); Roller et al. (2018); Yu et al. (2020). Others have used traditional word embedding, such as FastText Wang et al. (2019a); Sholikah et al. (2022); Jana et al. (2022) and Skip-gram with negative sampling Rei et al. (2018); Nguyen et al. (2017). GloVe Pennington et al. (2014) is one of the earliest representations that create word embedding based on a co-occurrence matrix of words on a specific corpus. In contrast, BERT **?** is one of the popular contextual embeddings, which, unlike traditional word embedding, gives a different embedding for a word based on its context. So the word bank will be given a different embedding if it appears in a financial context, i.e., a national bank, or in a natural context, i.e., revier Bank. Recent studies have proposed hypernymy-specific representations with the aim of modeling hypernymy relations efficiently. Several types of embedding are proposed; some studies have proposed hypernymy-specific neural word embedding Glavas & Vulic (2018); Yin & Roth (2018); Tan et al. (2020), others have proposed graph-based Wang et al. (2018); Liu et al. (2021) and geometric-based representations Tifrea et al. (2018); Nickel & Kiela (2017); Li et al. (2018); Wang et al. (2019b); Iwamoto et al. (2021). Poincare GloVe embedding is a type of geometric-based representation proposed by Tifrea et al., 2018. It represents words in the cartesian product of hyperbolic spaces, which is mapped to Gaussian embedding. The distance between the two word embeddings is the Fisher distance between their probability distribution function. The embeddings are learned using a generalized Glove method. It differs from the original Glove embedding Pennington et al. (2014), which is based on Euclidian space; the learning is adapted to hyperbolic space by editing the loss function. The embedding is evaluated on word similarity, analogy, and hypernymy detection. for hypernymy detection, it was trained on Levy and Goldberg corpus Levy et al. (2015) extracted from Wikipedia. They have compared the Poincare Glove embeddings with Vanilla Glove embeddings. They have found that Poincare Glove embedding outperforms Vanilla Glove embedding and that the initialization values benefit both embeddings. Moreover, the model trained using 50x2 dimensions Poincare balls outperforms others on the hypernymy detection task. Nickel & Kiela, 2017 have proposed a hyperbolic embedding based on a Poincare ball to represent hierarchical data. They have computed the embedding based on Riemannian optimization. The embedding is initialized randomly and trained on WordNet transitive closure; It was evaluated on taxonomy embeddings, link prediction tasks, and graded lexical entailment, which measures the degree of hypernymy relation between two terms Vulić et al. (2017). The result shows that it outperforms state-of-the-art embedding on lexical entailment.

A type of hypernymy-specific representation is the post-processed representations, which take pre-trained embedding as input and modify it to better represent hypernymy or other semantic relations by using semantic relations examples extracted from semantic resources such as wordnet Miller et al. (1990). One of the post-processing techniques is retrofitting. Vulić & Mrkšić, 2017 have proposed a retrofitting representation for lexical entailment called Lexical Entailment Attract-Repel (LEAR). Their retrofitting technique combines symmetric and asymmetric objectives. The symmetric objectives attract synonyms words vectors norm beside each other and repel antonyms words vectors norm far from each other. The asymmetric objectives attract the vector norm of lexical entailment words beside each other and enforce a hierarchal order for vector norms. Thus, the hypernym will have larger vector norms than hyponyms. They have used Skip-gram with negative sampling, Fastext, Context2Vec, and Glove embedding as input to LEAR. WordNet is the source of hypernymy, antonymy, and synonymy constraints used for training. LEAR was evaluated on hypernymy detection, hypernymy directionality, hypernymy detection, and directionality, and on graded lexical entailment. LEAR Embedding is able to outperform state-of-the-art models on all of these tasks. Nevertheless, it is limited by the availability of linguistics constraints.

Glavaš & Vulic, 2019 have proposed a Generalized Lexical ENtailment embedding model (GLEN) that learns a generalized lexical entailment function from lexical constraints, i.e., hypernymy, synonymy, and antonymy. The model can be applied to words with no known lexical constraint and generate proper embedding for them. It was evaluated on graded lexical entailment and cross-lingual hypernymy detection. The embedding model combines the benefits of retrofitting model and joint objectives models. They have created three learning objectives utilizing asymmetric Euclidian norms, symmetric cosine functions, and a regularization function to keep the useful information in the embedding space for each linguistic constraint type, lexical entailment, synonym, and antonym. In testing, they have combined the asymmetric and the symmetric functions to predict lexical entailment and graded lexical entailment. They have used pre-trained FastText embedding to learn the generalized lexical entailment function. The semantic constraints are collected from WordNet and Roget Thesaurus. They have compared the GLEN to the LEAR model for graded lexical entailment. They have found that GLEN is powerful on graded lexical entailment when fewer constraints are known, and it underperforms LEAR when more constraints are known. Moreover, there is a trade-off between generalizing for unseen constraints and the performance for seen constraints. GLEN is not limited by the availability of lexical constraints, and it could be helpful when a lot of unseen words are available. In our study, we will evaluate traditional word embedding and contextual word embedding against hypernymy-specific embedding. We will evaluate the effectiveness of GloVe, LEAR, GLEN, Poincare GloVe, Poincare embedding, and BERT in specific.

## 3 METHODOLOGY

The goal of our study is to evaluate the effectiveness of three types of representations on hypernym-related tasks. Furthermore, we aim to test if hypernymy-specific embedding is better at modeling hypernymy relation in the context of hypernymy-related tasks. The selected hypernymy-related tasks are hypernymy detection, hypernymy directionality detection, and semantic relation classification. To conduct the evaluation experiments, we have selected GloVe embedding as the traditional embedding baseline and BERT as the contextual embedding. For hypernymy-specific embedding, we have selected two retrofitted embeddings, LEAR and GLEN, and two geometrical-based embeddings, Poincare for hierarchical data and Poincare Glove. We have trained the embeddings on the AraBERT corpus Antoun et al.. We have used several datasets to train and test all these models. To mitigate external effects on the performance, we have tried to control most of the models' hyperparameters and the experimental setups. In the following subsections, we highlight the details of embedding training corpus, datasets, classification models, experimental setup, and hypernymy-related tasks.

### 3.1 CORPUS AND DATASETS

**AraBERT corpus:** We have trained all word embedding on the corpus used to train an Arabic version of BERT called AraBERT Antoun et al.. AraBERT is trained Arabic text extracted from the Arabic Wikipedia, The 1.5B words Arabic Corpus El-Khair (2016), unshuffled and filtered OSCAR corpus [1], The OSIAN Corpus Zeroual et al. (2019), and Assafir news articles [2]. The data size is 77GB, and the vocabulary is 12+ million. Training word embedding imposes multiple challenges; the demand for resources is very high, and the training needs large-size RAM, free disk space, and very efficient GPU. Moreover, the code of some of the embeddings is capable only of handling fewer words and takes more training time. Therefore, we have used half of the AraBERT corpus for training embeddings except for AraBERT embedding, which was pre-trained on the full corpus. To create AraBERT half corpus, we randomly selected 2006 files having 9+ million vocab and 38GB data size.

**Arabic Semantic Relation Dataset (ASRD):** We have created our in-house dataset for Arabic semantic relationships. The used version of the dataset contains one-word examples for hypernym, hyponym, has_instance, is_instance, entailment, synonym, meronym, holonym, attribute, antonym, cause, similar, and verb_group. The number of examples in ASRD(one) is 958341; the dataset statistics are presented on table 1. The dataset is extracted from multiple Arabic semantic resources;

---

[1] https://oscar-project.org/
[2] https://assafirarabi.com/en/

The Arabic wordnet[3] Elkateb et al. (2006); Abouenour et al. (2013), Open multilingual wordnet[4] Bond & Foster (2013), RADIF dictionary for antonyms and synonyms [5], The Arabic Ontology[6] Jarrar (2021), and The Qurann ontology Hakkoum & Raghay (2016). The dataset is split into 60%, 20%, and 20% for training, validation, and testing sets, respectively. ARSD datasets will be publicly available.

| Relation | Number of examples |
|---|---|
| hyponym | 368599 |
| hypernym | 365344 |
| synonym | 188225 |
| verb_groups | 16272 |
| entailments | 7734 |
| antonym | 27 |
| causes | 5022 |
| has_instance | 1447 |
| is_instance | 1447 |
| part_meronyms | 1583 |
| part_holonyms | 1575 |
| attributes | 474 |
| similar | 326 |
| also_sees | 266 |
| **Total** | **958341** |

Table 1: ASRD(one) Statistics

### 3.1.1 REPRESENTATIONS TRAINING

All the selected representations are trained on half of the AraBERT corpus mentioned above except Poincare, which is trained on ASRD hypernymy pairs, and BERT, which is pre-trained on the full AraBERT corpus. Following we will describe the training process of each embedding.

- **GLOVE:** We have trained GloVe on half of the AraBERT corpus after preprocessing it and combining it on one file in which each line is a document. First, the corpus is preprocessed using AraBERT preprocessor [7] to remove emojis, HTML markup, diacritics, letters elongation, and repetition and to replace Uniform Resource Locators (URLs), emails, and Hindi numerals. Furthermore, punctuation and English letters are removed. Finally, numbers are replaced with a special token. We have used the original GloVe code[8] to train our version with the setup mentioned in table 2.

| GloVe Settings | Value |
|---|---|
| Embedding dimensions | 100 |
| Iterations | 100 |
| Window size | 15 |
| Minimum count | 5 |
| Number of thread | 48 |
| Maximum memory | 110 GB |
| Machine | Machine 2 |

Table 2: GloVe Settings

---

[3]http://globalwordnet.org/resources/arabic-wordnet/
[4]https://omwn.org/omw1.html
[5]https://github.com/mdanok/arabicLTcontributing
[6]https://ontology.birzeit.edu/
[7]https://github.com/aub-mind/arabert
[8]https://github.com/stanfordnlp/GloVe/

- **LEAR:** LEAR is retrofitting-based embedding that takes pre-trained embeddings as input and modifies the embedding according to lexical-semantic relations constraints to better represent the relation. We have trained LEAR by using GloVe embedding mentioned above and lexical-semantic constraints extracted from ASRD. LEAR needs synonyms and hypernyms for its Attract objective and antonym for its repel objective. we have used 11979 antonyms, 368489 hypernyms, and 196054 synonyms with keeping duplicated examples. We have used the official python implementation of LEAR[9] with slight modifications to adapt it to our data and the newer version of Python. We have trained 100 dimensions embedding and tried 5, 20, and 100 iterations; see table 3 for LEAR training settings.

| LEAR Settings | Value |
| --- | --- |
| Embedding dimensions | 100 |
| Iterations | 5, 20, 100 |
| Pre-trained input | GloVe 100d |
| Machine | Machine 2 |

Table 3: LEAR Settings

- **GLEN:** GLEN takes a pre-trained embedding and lexical-semantic constraints as inputs and generates a generalized modified embedding for all vocabulary, even the one with no constraints. We have used GloVe embedding mentioned above and lexical-semantic constraints extracted from ASRD. For training, we have used 362684 hypernyms, 186587 synonyms, and 27 antonyms. Meanwhile, for development, we have used 678 hypernyms and 349 synonyms. We have used the official implementation of GLEN [10] with 100 dimensions and the default hyperparameters except for the number of iterations to stop training if there is no improvement on the development set (Table 4).

| GLEN Settings | Value |
| --- | --- |
| Embedding dimensions | 100 |
| Stop after iteration | 200 |
| MLP layers | 5 |
| Pre-trained input | GloVe 100d |
| Machine | Machine 1 |

Table 4: GLEN Settings

- **Poincare GloVe:** Poincare GloVe used a modified GloVe objective to generate new word embedding. It does not necessarily use pre-trained word embedding or lexical-semantic constraints; rather, we have used the co-occurrence calculation file generated by our GloVe training as a basis for its calculation. We have trained two versions of Poincare GloVe, 100D Poincare GloVe using all vocabulary and $cosh^2$ as the distance function trained using 100D Poincare ball, and $50 \times 2D$ Poincare GloVe that uses the most frequent 539642 words of the vocabulary and $x^2$ distance function and trained in the cartesian product of 50 2D Poincare balls. Tabel5 shows Poincare training set-up.

| 100D Poincare GloVe | Value | 50×2D Poincare GloVe | Value |
| --- | --- | --- | --- |
| Embedding dimensions | 100 | Embedding dimensions | 50×2 |
| Iterations | 50 | Iterations | 23 |
| Optimization | RadaGrad | Optimization | Mix RadaGrad |
| Learning rate | 0.01 | Learning rate | 0.05 |
| Machine | Machine 1 | Machine | Machine 1 |

Table 5: 100D and 50×2D Poincare GloVe settings

---

[9]https://github.com/nmrksic/LEAR
[10]https://github.com/codogogo/glen

- **Poincare Embedding:** The Poincare embedding is trained using lexical-semantic constraints with a tree-like structure. In our training, we use 366791 constraints extracted from ASRD for hypernym and has_instance examples. The number of negative examples is set to 5. The resulting embedding is 50 dimensions. We have used Gensim implementation of Poincare embedding[11].

- **BERT:** BERT is a contextual embedding pre-trained on a corpus. In our evaluation, we do not retrain BERT; for each term, we have extracted features of the final layer output from pre-trained AraBERT V2. Before that, we prepared the input terms, converted them to tokens, converted tokens to IDs, and created a token tensor and segment tensor. Table 6 shows AraBERT features extraction settings.

| AraBERT Settings | Value |
|---|---|
| Model | aubmindlab/bert-base-arabertv02 |
| Tokenizer | aubmindlab/bert-base-arabertv02 |
| Features | pooler_output |
| Embedding dimensions | 768 |

Table 6: AraBERT Extraction Settings

### 3.1.2 CLASSIFICATION MODELS AND TASKS

To Assess the effectiveness of the chosen representations in modeling hypernymy relations, we have used the resulting embeddings from each model as input to three hypernymy-related tasks: hypernymy detection, hypernymy directionality, and semantic relation classification. The goal of our evaluation was not to achieve the highest performance but rather to fairly evaluate representation models by keeping experiment variables consistent among different experiments. Thus, for each task, we have used a simple feed-forward neural classification model with an embedding layer, one hidden layer, and an output layer. The tasks differed in the number of output targets and dataset sizes based on the type of relations involved in the task. We have trained a model for each embedding on each task. For evaluating the classification models, we test the trained model on several datasets, including the test set of ASRD. Following, we will describe each classification model.

- **Hypernymy detection:** The detection model will classify input examples as hypernymy or not, leading to two classes in the output layer. The complete ASRD datasets are used to train, tune, and evaluate the model. ASRD positive examples are hypernyms, entailment, and has_instance; other relations are considered negative examples.

- **Hypernymy directionality detection:** The directionality detection model determines the direction of the relation by classifying examples into two categories: hypernymy or hyponymy. For this task, Only hypernyms, has_instance, hyponyms, and is_instance from ASRD are used .

- **Semantic relation classification:** The SRC model will classify a number of lexical-semantic relations, including hypernymy. For this task, we have trained two models for each embedding, each with a different set of relations. The first considers hypernymy, meronomy, synonymy, antonymy, and attribute. While the second considers hypernymy, synonymy, and autonomy.

### 3.1.3 EVALUATION DATASETS

We utilized lexical-semantic constraints extracted from the ASRD to train both the embedding models and the classification models. This suggests that a shared vocabulary might influence the performance of the embeddings. To mitigate this effect, we have used eight datasets other than ASRD with varying characteristics. Specifically, we have selected eight English benchmark datasets containing hypernymy relations[12] and translated them into Arabic using Google Translate[13]. Additionally, We

---

[11]https://radimrehurek.com/gensim$_3$.8.3/$models/poincare.html$

[12]https://github.com/ahug/HypEval/tree/master/data

[13]https://translate.google.com

have filtered the terms in these datasets to include only single-word entries that were present in the training corpus. Here we briefly describe these datasets.

- **BLESS Baroni & Lenci (2011):** It contains 200 single-word living and non-living concepts linked with five relations to more than 26,000 relata with different part-of-speech tags. The presented relations are coordination (i.e., co-hyponymy), hypernymy, meronymy, concept attribute, related event, and random.

- **BIBLESS Kiela et al. (2015):** Contains a relabeled version of Weeds dataset. The hypernymy pairs are labeled with 1, hyponymy pairs with -1, and other pairs with 0.

- **ENTAILMENT Baroni et al. (2012):** It is a dataset dedicated to entailment among multi-word expressions and single words.

- **EVALution Santus et al. (2015):** It is a large dataset extracted from WordNet and ConceptNet and filtered using automatic techniques and human judgments

- **Lenci/Benotto Lenci & Benotto (2012):** a BLESS subset dataset which extract hypernymy and hyponymy from BLESS

- **Weeds Weeds et al. (2014)** This version of WBless contains 2929 hypernym and co-hypoynym examples.

- **Root9 Santus et al. (2016):** It is a dataset created by extracting random pairs of hypernymy, co-hyponymy, and random words in different part-of-speech from EVALution, Lenci/Benotto, and BLESS.

### 3.1.4 EXPERMINTAL SET-UP

In the previous section, we highlight the setup of representation training. In this section, we describe the setup of the neural classification models, the training settings, and the computing machine. For the neural classification models of hypernymy-related tasks, we have concatenated the embeddings of both input terms, leading to an input layer with a size equal to 2xembedding dimensions. We use cross-entropy loss, Stochastic Gradient Descent (SGD) optimizer, 150 dimensions hidden layer, and ReLU activation function on the output layer. We have trained each model for 50 epochs except for models that use BERT representation and tested only on ASRD for hypernymy detection task due to time and computing power limitations. The Hypernymy Detection model with BERT is trained for 43 epochs. Unless otherwise mentioned, each of the embeddings is trained with the authors' default setting. Due to time and computing power limitations, the 50x2D Poincare embedding is trained with 23 epochs instead of 50. In order to evaluate the effect of the selected representations, we have tried to control the esp All of our expermints are conducted on two Machines; *Machine 1* with Nvidia RTX GeForce 4080 with 16GB RAM GPU, 13th Gen Intel(R) Core(TM) i7-13700k CPU, 135GB RAM, and 3.7 TB disk space. *Machine 2* with NVIDIA GeForce RTX 4090 24GB RAM GPU, 13th Gen Intel(R) Core(TM) i9-13900K CPU, 125GB RAM and 1.8TB disk space.

## 4 RESULTS AND DISCUSSION

### 4.1 HYPERNYMY DIRECTIONALITY DETECTION

Tabel 7 shows the experiment results for the 8 embeddings on 8 datasets. On ASRD, the best-performing model is 100D Poincare GloVe, but 50x2D Poincare GloVe, LEAR 5, LEAR 20, Poincare embedding, and GLEN performed relatively similarly. GloVe baseline and LEAR 100 are the least-performing models.

Comparing LEAR versions with the GloVE baseline shows similar performance in most cases except on ASRD. This indicates that LEAR is powerful if it is trained on constraints similar to the dataset and less useful when fewer constraints are available; it performs similarly to GloVe or sometimes less. Similarly, Poincare GloVe performs better than the GloVe baseline when the constraints are similar to the dataset and perform similarly to GloVE otherwise. In the case of GLEN embedding, it outperforms GloVe on all datasets except BLESS and Root_a. Furthermore, it outperforms other embedding on 3 evaluation datasets, which is consistent with Glavaš & Vulic, 2019 findings that GLEN is better when fewer constraints are known.

On 6 out of 8 datasets, 100D Poincare GloVe is the best-performing embedding for the detection tasks, followed by GLEN on 3 out of 8 datasets. The results show that 100D Poincare GloVe is the best embedding on the Directionality detection task.

| Dataset | #Examp | GloVe | Lear 5 | LEAR 20 | LEAR 100 | Poincare Embedding | 100D Poincare GloVe | 50x2D Poincare GloVe | GLEN |
|---------|--------|-------|--------|---------|----------|--------------------|---------------------|----------------------|------|
| ASRD    | 147368 | 0.81  | 0.85   | 0.85    | 0.84     | 0.89               | **0.88**            | 0.85                 | 0.86 |
| BLESS   | 10409  | 0.45  | 0.42   | 0.41    | 0.41     | 0.44               | **0.49**            | 0.48                 | 0.41 |
| BIBLE.  | 1743   | 0.71  | 0.71   | 0.66    | 0.66     | 0.64               | 0.64                | 0.66                 | **0.73** |
| ENTA.   | 2755   | 0.67  | 0.67   | 0.64    | 0.66     | 0.59               | 0.68                | 0.67                 | **0.71** |
| L&B     | 4604   | 0.55  | 0.55   | 0.54    | 0.53     | 0.51               | **0.57**            | 0.56                 | 0.56 |
| Weeds   | 3128   | **0.60** | 0.59 | 0.57    | 0.56     | 0.55               | **0.60**            | **0.60**             | **0.60** |
| Root9_a | 8139   | 0.55  | 0.53   | 0.52    | 0.51     | 0.53               | 0.58                | 0.58                 | 0.54 |
| Root9_b | 11728  | 0.50  | 0.48   | 0.48    | 0.47     | 0.49               | **0.53**            | 0.52                 | 0.49 |

Table 7: F1-score results for the directionality detection task (BIBLE. is BIBLESS, ENTAI. is ENTAILMENT, and L&B is LenciBenotto)

## 4.2 HYPERNYMY DETECTION

Tabel 8 shows the results of evaluating different embeddings on the hypernymy detection task on 8 datasets. On ASRD, the best-performing model is Poincare embedding, followed by 50x2D Poincare GloVe. Moreover, all hypernymy-specific embeddings outperform the GloVe baseline. On 4 out of 8 datasets, Poincare GloVe models outperform others, Followed by GLEN, which outperforms other embeddings on 2 out of 8 datasets and performs similarly to the best embedding model on the other 4 datasets. 100D Poincare Glove outperforms other embeddings on BLESS, Root9_a, and Root9_b, while 50x2D Poincare GloVe outperforms others on the Weeds dataset. GLEN has the best performance on BIBLESS and ENTAILMENT and has a similar performance to the best model on ASRD, Lenci/Benotto, Weeds, and Root9_a datasets. Surprisingly, BERT is the least-performing model on the ASRD dataset, and it is outperformed by all hypernymy-specific embeddings, which might indicate the difficulty of the task. LEAR performs similarly to the GloVe baseline on all datasets, and Poincare embedding performs lower than the GloVe baseline on all datasets except ASRD, which is reasonable since it was trained on ASRD lexical-semantic constraints. The results show that the choice of the datasets plays a major role in the performance of the hypernymy detection task.

| Dataset | #Examples | GloVe | Lear 5 | LEAR 20 | LEAR 100 | Poincare Embedding | 100D Poincare GloVe | 50x2D Poincare GloVe | GLEN | BERT |
|---------|-----------|-------|--------|---------|----------|--------------------|---------------------|----------------------|------|------|
| ASRD    | 191668    | 0.76  | 0.80   | 0.80    | 0.80     | **0.84**           | 0.83                | 0.80                 | 0.81 | 0.62 |
| BLESS   | 9486      | 0.57  | 0.52   | 0.53    | 0.53     | 0.55               | **0.58**            | 0.53                 | 0.51 | NA   |
| BIBLE.  | 1167      | 0.67  | 0.69   | 0.67    | 0.65     | 0.48               | 0.63                | 0.64                 | **0.72** | NA |
| ENTAI.  | 1837      | 0.64  | 0.64   | 0.62    | 0.62     | 0.51               | 0.65                | 0.64                 | **0.66** | NA |
| L&B     | 3253      | 0.55  | **0.57** | 0.55  | 0.54     | 0.44               | 0.53                | 0.56                 | 0.55 | NA   |
| Weeds   | 2074      | **0.58** | **0.58** | 0.56 | 0.55    | 0.46               | 0.57                | **0.58**             | 0.57 | NA   |
| Root9a  | 6181      | 0.59  | 0.54   | 0.54    | 0.55     | 0.50               | **0.61**            | 0.57                 | 0.58 | NA   |
| Root9b  | 9233      | 0.55  | 0.52   | 0.53    | 0.53     | 0.54               | **0.56**            | 0.54                 | 0.52 | NA   |

Table 8: F1-score results for the hypernymy detection task (BIBLE. is BIBLESS, ENTAI. is ENTAILMENT, and L&B is LenciBenotto)

## 4.3 SEMANTIC RELATION CLASSIFICATION

Tabel9 shows the result of using a different representation model on 3 datasets. We have filtered relations out of ASRD that are not available on the testing set and trained a separate model for each test set. The results are relatively low. This could be attributed to fewer examples representing each class, especially the autonomy class, which has only 27 examples. Nevertheless, GloVe is the least-performing embedding in all datasets except on Lenci/Benotto datasets Poincare GloVe is lower.

Poincare Embedding is the best-performing model on the ASRD test set, which is reasonable since it was trained on ASRD lexical-semantic constraints. Moreover, it is the best-performing model on the evaluation dataset. LEAR is the best-performing embedding on Lenci/Benotto datasets. Similar to hypernymy detection results, the results show that the datasets have an effect on the performance of the SRC task.

| Dataset | #Examples | GloVe | Lear 5 | LEAR 20 | LEAR 100 | Poincare Embedding | 100D Poincare GloVe | 50x2D Poincare GloVe | GLEN |
|---------|-----------|-------|--------|---------|----------|--------------------|---------------------|----------------------|------|
| ASRD | 112967 | 0.28 | 0.43 | 0.43 | 0.45 | **0.52** | 0.42 | 0.40 | 0.48 |
| Eval. | 12095 | 0.13 | 0.16 | 0.16 | 0.16 | **0.20** | 0.15 | 0.16 | 0.16 |
| ASRD | 112556 | 0.50 | 0.52 | 0.52 | 0.51 | **0.54** | 0.53 | 0.51 | 0.51 |
| L&B | 3253 | 0.28 | **0.31** | **0.31** | **0.31** | 0.19 | 0.29 | 0.30 | 0.30 |

Table 9: F1-score results for the Semantic relation classification task (Eval. is Evaluation and L&B is LenciBenotto)

## 4.4 DISCUSSION

From the results in the previous subsections, we observe that, despite being trained without lexical-semantics constraints, Poincare GloVe iconsistently performs best on the hypernymy detection and hypernymy directionality detection tasks. Additionally, 100D Poincare GloVe slightly outperforms its counterpart, 50x2D Poincare GloVe. This highlights the effectiveness of modeling hypernymy relation in hyperbolic space even without even being exposed to explicit hypernymy constraints. However, in the semantic relation classification task, which involves non-hierarchical relations, Poincare GloVe shows lower performance. This could be attributed to the fact that non-hierarchical relations are less suited to hyperbolic modeling. On the other hand, GLEN outperforms other representations on some of the hypernymy detection and directionality datasets, although it falls short on ASRD. This is expected given that ASRD constraints are used to train GLEN, and it has been known to have more impact when used with datasets with fewer known constraints Glavaš & Vulic, 2019.

The results also reveal that no single representation outperforms consistently outperforms the others across all evaluation datasets; different representations perform differently on different datasets. This suggests that the way training and evaluation datasets are constructed plays a crucial role in model performance. This observation is supported by the findings of Chang et al., 2017, which show that the way negative examples are constructed in the dataset has a significant impact on model performance.

## 5 CONCLUSION

In this work, we investigated the impact of various types of embeddings on the performance of three hypernymy-related tasks. We selected a diverse range of embeddings: one traditional neural word embedding (GloVe), one contextual word embedding (BERT), four hypernymy-specific embeddings, including two geometric-based embeddings (Poincare GloVe and Poincare embedding), and two retrofitting-based embeddings (LEAR and GLEN). These embeddings were trained on half of the AraBERT corpus. The effectvense of the embeddings was evaluated across three tasks: hypernymy detection, hypernymy directionality detection, and semantic relation classification. The classification models of these tasks were trained on our Arabic Semantic Relation Dataset (ASRD) and tested on the ASRD test set and eight translated English benchmarked datasets.

The experimental results demonstrate that Poincare GloVe can effectively model hypernymy relation, even ithout incorporating explicit constraints during training, while GLEN performs well on datasets with fewer known constraints. Furthermore, our findings suggest that the choice of dataset used in the training and evaluation has a significant effect on model performance. This highlights the importance of carefully designing datasets and selecting training examples.

Future work will include exploring additional hypernymy-specific embeddings, such as hierarchical and graph-based embeddings. We also plan to experiment with various unsupervised metrics proposed in the literature, including informativnesse and distributional inclusion hypothesis measures.

ACKNOWLEDGMENTS

The authors would like to acknowledge the support of X at Y through the grant Z.

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
