# OpenReview forum: "Evaluating word representation for hypernymy relation: with focus on Arabic"
_ICLR.cc/2025/Conference — ICLR 2025 Conference Withdrawn Submission_

### Official Review · Reviewer_DU1q · 2024-10-27

**Soundness:** 2
**Presentation:** 2
**Contribution:** 2
**Rating:** 3
**Confidence:** 4

**Summary:**

This paper addresses the challenge of modeling hypernymy relations specifically focusing on Arabic. The authors evaluate various word representation methods to determine the most effective for Arabic hypernymy tasks. They compare traditional embeddings, hypernymy-specific embeddings, and contextual embeddings using an Arabic corpus and multiple datasets to assess their impact on tasks such as hypernymy detection.

**Strengths:**

1. Word representations for hypernymy are essential for a variety of tasks in NLP and information extraction.

2. By concentrating on the Arabic language, the paper contributes to a less explored area, providing insights for non-English NLP research.

3. The research conducts an evaluation of multiple types of embeddings, including traditional, hypernymy-specific and contextual embeddings.

**Weaknesses:**

1. The paper primarily focuses on evaluating existing word representations rather than introducing a novel approach or method for hypernymy modeling. The novelty is very limited.

2. The written quality of this paper is poor. The authors should carefully revise this paper for better presentations. For example, the citation format is incorrect.

3. The paper seems to provide an evaluation of performance effects without a deep analysis of why certain embeddings perform better or worse in specific contexts or tasks.

4. The impact of this work in the ICLR community is limited. Maybe an NLP workshop on the Arabic language is more suitable.

**Questions:**

There are no specific questions here. But I suggest that the authors might consider proposing a novel method or modification in word representation tailored to hypernymy in Arabic instead of solely focusing on evaluation. In addition, this work may be better suited for a specialized venue, such as an NLP workshop focused on the Arabic language. This could provide a more appropriate audience that appreciates the linguistic specificity.

---

### Official Review · Reviewer_yyTA · 2024-10-30

**Soundness:** 2
**Presentation:** 2
**Contribution:** 1
**Rating:** 3
**Confidence:** 3

**Summary:**

This paper focuses on evaluating and improving word vector representations specialized for hypernym relations in Arabic. Your research captures the gaps in hypernym aspects of performance by conducting multiple sets of experiments on different datasets, and this aspect of the research is important for the NLP task.

**Strengths:**

1. **Relevance**: Focusing on hypernym in Arabic is timely and relevant, and it addresses a less explored area of NLP.
2. **Experiments**: The paper presents a comprehensive experimental evaluation of multiple word representation techniques, demonstrating a solid methodological framework.
3. **New findings**: The findings presented in the paper provide new insights into how Arabic word embeddings can be enhanced to better detect hypernym.

**Weaknesses:**

1. **Literature Review**: Although the paper discusses related work, a more comprehensive literature review would have positioned the paper's contribution more effectively. I suggest the authors report precision and recall specifically for elements that have both spatial and logical relationships, compared to those with only one type of relationship.
2. **Clarity**: Some sections may lack clarity, particularly in explaining the significance of the research methodology and findings.
3. **Results Interpretation**: The results section may need to discuss the significance of the findings in more depth, especially in the context of existing models.

**Questions:**

1. How do the results of your study compare to existing models in other languages, especially in terms of applicability to Arabic?
3. What are the advantages of your study over previous studies? Is a new methodology or a new dataset proposed?
### Additional feedback
- Consider revising the introduction to better outline the motivation for the study and its significance in the broader context of NLP research.
- It may be helpful to include more hypernym examples in Arabic throughout the paper to illustrate your points.

---

### Official Review · Reviewer_SeDp · 2024-11-04

**Soundness:** 3
**Presentation:** 1
**Contribution:** 2
**Rating:** 3
**Confidence:** 3

**Summary:**

Authors try to evaluate different algorithms, which create hypernymy relation representations in Arabic. They select AraBERT corpus as a base for training all embedding models and train several models on this data. As a baseline for contextual embeddings, BERT is used, while for classic embeddings GloVe is used. For hypernymy-specific embedding LEAR, GLEN, Princare and Poincare Glove is used. After that, a simple feedforward models are trained for all embeddings for three tasks: hypernymy detection, hypernymy directionality detection and semantic relation classification. Results show, that Poincare GloVe performs best on hypernomy detection and hypernomy directionality detection tasks. In semantic relation classification tasks Poincare GloVe performs worse. Overall, there is no best representation for all tasks.

**Strengths:**

1. The goal of the paper is easy to understand, as it provides valuable insight into which representations are best for hypernym-based tasks (none are best overall)

2. Experiment design makes sense and is mostly without issues. There is a minor issue stemming from the limited resources available to the authors of the paper, but I will touch upon them in the weaknesses part of the review.

**Weaknesses:**

1. Authors are very constrained in resources, having to resort to halving the size of the training dataset for some of the algorithms. This raises questions to the validity of the collected information, since Poincare GloVe, the best algorithm in Hypernymy Directionality and Hypernymy Detection tasks, has seen only half as many data samples, which can possibly make the results non-representative. However, due to the simplicity of the Poincare GloVe, most likely it won’t impact the results as much, thus, making this just a minor issue.

2. The quality of the text's presentation is poor; it contains numerous typos and improperly formatted tables. Table 7 has incorrectly formatted items in header, Table 8 has incorrectly formatted dataset names, in table 7 the highest F1 score for ASRD dataset is incorrectly attributed to 100D Poincare GloVe, which has the score of 0.88 instead of Poincare Embedding, which have the score of 0.89. On the line 070 BERT has no citation available, on the line 220 the sentence starts from lowercase, on the line 291 the word Assess is incorrectly capitalized, line 480 is cut in half, etc. Both Introduction and Related Work sections are hard to read, since they are written in a big wall of text instead of separate paragraphs on groups of algorithms. Some of the citation years are in brackets (lines 065-068), some are not (line 079).

**Questions:**

Please, correct the formatting of the paper, it is very hard to read in current condition.

---

### Note · Authors · 2024-11-26

**Comment:**

Dear reviewers,

I greatly appreciate the time and effort the reviewers have dedicated to providing valuable and insightful feedback, which has been instrumental in improving the quality of my work. I would like to withdraw the submission. I have addressed the reviewers' comments and made improvements to the paper. Your feedback has allowed me to re-write my contribution and make it more explicit. I have emphasized that, to the best of our knowledge, no prior work has conducted an empirical comparison between different types of representations, specifically hypernymy-specific embeddings. I have also clarified the unified process of training all representations on the same corpus and vocabulary, which shows the experiment's validity. I also made minor corrections to the results table to ensure accuracy and consistency.

Despite that, After the initial submission, I conducted two additional sub-experiments. These experiments significantly enrich the paper by providing more context, valuable results, and a broader contribution to the field. The paper now has necessitated changes that extend beyond minor revisions.

While I have benefited from the feedback, withdrawing the paper now is more appropriate to ensure that the updated version reflects the full scope of the research.

Thank you for understanding. I hope to resubmit a more comprehensive version of this work to a future venue.

Regards,

**Withdrawal Confirmation:**

I have read and agree with the venue's withdrawal policy on behalf of myself and my co-authors.